# When Is Inductive Inference Possible?

**Zhou Lu**
Princeton University
zhoul@princeton.edu

## Abstract

Can a physicist make only a finite number of errors in the eternal quest to uncover the law of nature? This millennium-old philosophical problem, known as inductive inference, lies at the heart of epistemology. Despite its significance to understanding human reasoning, a rigorous justification of inductive inference has remained elusive. At a high level, inductive inference asks whether one can make at most finite errors amidst an infinite sequence of observations, when deducing the correct hypothesis from a given hypothesis class. Historically, the only theoretical guarantee has been that if the hypothesis class is countable, inductive inference is possible, as exemplified by Solomonoff induction for learning Turing machines. In this paper, we provide a tight characterization of inductive inference by establishing a novel link to online learning theory. As our main result, we prove that inductive inference is possible if and only if the hypothesis class is a countable union of online learnable classes, potentially with an uncountable size, no matter the observations are adaptively chosen or iid sampled. Moreover, the same condition is also sufficient and necessary in the agnostic setting, where any hypothesis class meeting this criterion enjoys an $\tilde{O}(\sqrt{T})$ regret bound for any time step $T$, while others require an arbitrarily slow rate of regret. Our main technical tool is a novel non-uniform online learning framework, which may be of independent interest.

## 1 Introduction

How does one move from observations to scientific laws? The question of induction, one of the oldest and most basic problems in philosophy, dates back to the 4th century BCE when Aristotle recognized deduction and induction as the cornerstones of human reasoning. Unlike deduction, where conclusions are drawn logically from premises, induction extrapolates general principles from empirical observations without such certainty.

The justification of induction has historically been contentious. David Hume, in his 'Problem of Induction', argued that extrapolations based on past experiences cannot reliably predict the unexperienced. Despite such skepticism, induction remains essential for advancing from specific instances to general theories, deriving first principles upon which rigorous deductive reasoning is built, as Aristotle put it in 'Nicomachean Ethics':

> *"Induction is the starting-point which knowledge even of the universal presupposes, while syllogism proceeds from the universals."*

This highlights induction's crucial role in human reasoning and scientific discovery. Yet, the enduring question remains: how do we model induction, and what guarantees can it offer?

Inductive inference Gold [1967], as the most representative mathematical abstraction of induction, provides a rigorous framework for modeling induction. In inductive inference, a learner aims to deduce the ground-truth hypothesis $h^*$ from a hypothesis class $\mathcal{H}$ based on an infinite observation sequence $\{x_t\}$. At each round $t$, the learner makes a binary prediction $y_t$ based on all previous

information after observing $x_t$, then the true outcome $h^*(x_t)$ is revealed and the learner makes an error if $y_t \neq h^*(x_t)$. For example, the hypothesis class $\mathcal{H}$ can represent different physical models, and the observations are the data collected by a physicist, who refines theories based on new evidence.

**The question of inductive inference is whether the learner can make a finite number of errors depending on $h^*$ in such infinite process.** Unfortunately, despite its critical importance to philosophy, theories for inductive inference are notably sparse. All existing theoretical results, including the famous Solomonoff induction Solomonoff [1964a] for learning Turing machines, provide only a sufficient condition on the size of the hypothesis class: *inductive inference is possible, if $|\mathcal{H}| \leq \aleph_0$,* i.e. the size of the hypothesis class $\mathcal{H}$ is at most countable.

Clearly $|\mathcal{H}| \leq \aleph_0$ is not a necessary condition. Consider $\mathcal{H} = \{f_c | f_c(x) = 1_{x=c}, c \in \mathbb{R}\}$, it has an uncountable size but can be easily learnt with at most one error: the learner keeps predicting $y_t = 0$ until an error is made, then $h^*$ can be identified. In this work, we provide a sufficient and necessary condition for inductive inference, via a novel link to online learning theory. Our main result is the following (as a consequence of Theorems 9, 11):

**Theorem 1.** *Inductive inference is possible, if and only if $\mathcal{H} = \cup_{n \in \mathbb{N}} \mathcal{H}_n$, where each $\mathcal{H}_n$ is an online learnable class (a hypothesis class with finite Littlestone dimension).*

As an extension, we further demonstrate that the condition $\mathcal{H}$ being a countable union of online learnable classes is also sufficient and necessary for the agnostic setting, where the ground-truth $h^*$ may not lie in $\mathcal{H}$. For the weaker criterion of consistency where the error bound additionally depends on the observation sequence, we derive a necessary condition which we conjecture to be tight. Algorithms from previous works, examples, and technical proofs are relegated to the appendix.

## 1.1 Our Approach

Due to the shared spirit of sequential decision-making between classic online learning theory Littlestone [1988] and inductive inference Solomonoff [1964a], we seek to cast the problem of inductive inference within the online learning paradigm, which can handle uncountable-sized and agnostic hypothesis classes. The challenge is that traditional online learning theory itself does not fully capture the nuances of inductive inference, for its focus on an adaptive adversary and worst-case uniform error bounds.

To bridge the gap between inductive inference and online learning, we propose a new framework termed non-uniform online learning to bypass these differences. Different from classic online learning, this setting considers an oblivious choice of the ground-truth hypothesis (Nature can't change her mind on the choice of $h^*$) and non-uniform guarantees (error bound can vary for different $h^*$). It can be seen as a natural generalization of the non-uniform PAC learning notion Benedek and Itai [1988] to the online setting. The comparison between the protocols of classic and non-uniform online learning is made below.

| Classic online learning | Non-uniform online learning (inductive inference) |
|---|---|
| 1: Given domain $\mathcal{X}$ and hypothesis class $\mathcal{H}$ | 1: Given domain $\mathcal{X}$ and hypothesis class $\mathcal{H}$ |
| 2: | 2: **Nature selects ground-truth $h^* \in \mathcal{H}$** |
| 3: **for** $t = 1, \ldots, \infty$ **do** | 3: **for** $t = 1, \ldots, \infty$ **do** |
| 4:    Nature presents observation $x_t$ to Learner | 4:    Nature presents observation $x_t$ to Learner |
| 5:    Learner predicts $y_t$ | 5:    Learner predicts $y_t$ |
| 6:    **Nature selects a consistent $h_t \in \mathcal{H}$** | 6: |
| 7:    Nature reveals the true label $h_t(x_t)$ | 7:    Nature reveals the true label $h^*(x_t)$ |
| 8: **end for** | 8: **end for** |
| 9: **Goal: a uniform error bound** | 9: **Goal: error bound can depend on $h^*$** |

In the realizable setting, non-uniform online learning is indeed **equivalent to inductive inference**, containing previous results as special cases. We show that Learner can achieve a finite error bound dependent on $h^*$, if and only if $\mathcal{H}$ is a countable union of hypothesis classes with finite Littlestone dimensions. Furthermore, for the weaker criterion of consistency which only requires a (non-hypothesis-wise) finite error bound, we obtain a necessary condition that we conjecture to be tight.

In the agnostic setting, when $\mathcal{H}$ is expressed as $\cup_{n \in \mathbb{N}^+} \mathcal{H}_n$ where each $\mathcal{H}_n$ has finite Littlestone dimension $d_n$, we propose an algorithm with an $\tilde{O}(\sqrt{d_n T})$ regret bound for any time step $T$ when $h^*$ lies in $\mathcal{H}_n$. In addition, the sharpness of such characterization is demonstrated by a trichotomy of regret's dependency on $T$ through the two complementary facts: (1) as long as $\mathcal{H}$ can't be written in this form, it requires arbitrarily slow rates (2) any non-trivial $\mathcal{H}$ requires $\Omega(\sqrt{T})$ regret.

## 1.2 Related Works

**Inductive inference and learning**   The pioneer works of Solomonoff Solomonoff [1964a,b, 1978] established a rigorous theoretical framework for inductive inference. Via a Bayesian approach, Solomonoff's method of prediction assigns larger prior weights to hypotheses with shorter description lengths, providing finite-error guarantees based on the prior weight of the ground truth. The principle of Occam's Razor that simplicity leads to better learning guarantees, as the cornerstone of Solomonoff's theory, plays a vital role in the evolution of statistical learning Blumer et al. [1987, 1989], Mitchell [1997], Von Luxburg and Schölkopf [2011], Harman and Kulkarni [2012], Shalev-Shwartz and Ben-David [2014]. VC dimension Vapnik and Chervonenkis [2015], the fundamental concept in PAC learning, is widely regarded as a measure of simplicity in learning theory Sterkenburg and Grünwald [2021], Sterkenburg [2023]. Besides theoretical investigations, there is also growing interest in the empirical inductive inference capabilities of large language models Wang et al. [2023], Qiu et al. [2023], Honovich et al. [2022], Gendron et al. [2023], Grau-Moya et al. [2024].

**Online learning**   Littlestone's seminal work Littlestone [1988] on online learnability in the realizable case introduced the Littlestone dimension, a combinatorial measure precisely characterizing the number of errors. This concept was extended to the learning with expert advice setting, where the weighted majority algorithm Littlestone and Warmuth [1994] achieves an $\tilde{O}(\sqrt{T})$ regret. Since then, online learning has expanded into diverse scenarios, including the bandit setting Auer et al. [1995], the agnostic setting Ben-David et al. [2009], the changing comparator setting Herbster and Warmuth [1998], and the convex optimization area Zinkevich [2003], Kalai and Vempala [2005], Hazan et al. [2007], Duchi et al. [2011], Hazan et al. [2016].

**Non-uniform learning**   Though less explored, the non-uniform learning framework Benedek and Itai [1988] which allows hypothesis-dependent generalization bounds, is crucial for our study. For a comprehensive introduction on non-uniform learning see Shalev-Shwartz and Ben-David [2014]. Closely related to our work in non-uniform learning are Wu and Santhanam [2021] and Bousquet et al. [2021]. The work of Wu and Santhanam [2021] considered the setting they named EAS (eventually almost surely), which demands finite error bounds with probability 1, when the data stream is iid samples from some known distribution. The work of Bousquet et al. [2021] considers how fast the generalization error decreases as the size of training data increases, based on a new combinatorial structure called the VCL tree. In addition to inductive bias on the hypothesis class, there is extensive research on universal consistency under stochastic process assumptions on observations Stone [1977], Antos and Lugosi [1996], Hanneke et al. [2020], Hanneke [2021], Blanchard et al. [2022], Blanchard [2022]. These works on universal learning typically utilize a $k$-nearest neighbor type algorithm for the constructive proof, which doesn't fully align with practical scenarios of human reasoning and scientific discovery.

## 2   Setup

Given an infinite domain $\mathcal{X}$ and the set of all $0/1$ loss functions on $\mathcal{X}$: $\mathcal{S} = 2^{\mathcal{X}}$, we consider a non-empty subset $\mathcal{H} \subset \mathcal{S}$ as the hypothesis class. Both $\mathcal{X}$ and $\mathcal{H}$ are known to Learner and Nature in advance. In inductive inference, Nature iteratively presents Learner with $x_t \in \mathcal{X}$ and asks Learner to predict $\hat{y}_t \in \{0, 1\}$, then reveals the the true label $y_t \in \{0, 1\}$ to Learner, while Learner aims to make as few errors as possible by inferring $\mathcal{H}$.

Formally, we consider a learning algorithm $\mathcal{A}$ of Learner, as any function of historical observations (thus the learning algorithm is adaptive). In particular, a deterministic learning algorithm is any function with the following domain and codomain

$$\cup_{n \in \mathbb{N}} \left( \Pi_{i=1}^n (\mathcal{X} \times \{0, 1\}) \times \mathcal{X} \right) \rightarrow \{0, 1\}.$$

A randomized learning algorithm is similarly defined by setting the codomain as $[0, 1]$.

Throughout this paper, we will work with the following general sequential game between Learner and Nature. Before the game begins, Learner fixes a learning algorithm $\mathcal{A}$. At time step $t \in \mathbb{N}^+$, some $x_t$ is presented by Nature to Learner, and Learner predicts

$$\hat{y}_t = \mathcal{A}(\Pi_{i=1}^{t-1}(x_i \times y_i) \times \{x_t\}).$$

Then Nature reveals the true label $y_t$ and Learner makes an error if $\hat{y}_t \neq y_t$. We write $x = \Pi_{i \in \mathbb{N}^+}\{x_i\}$ and the set of all possible $x$ as $X$. For simplicity, we assume the continuum hypothesis, i.e. $2^{\aleph_0} = \aleph_1$.

In the following, we discuss different ways of Nature choosing $(x_t, y_t)$ and the criteria of learning, corresponding to different notions of learnability.

## 2.1 Classic Online Learning

In the classic online learning setting, there is no restriction on Nature's choice of $x_t$, and the only restriction on $y_t$ is the existence of a hypothesis $h_t^* \in \mathcal{H}$ consistent with history: $\forall i \leq t, h_t^*(x_i) = y_i$. Classic online learnability is characterized by a combinatorial quantity of the hypothesis class $\mathcal{H}$, called the Littlestone dimension $\mathbf{Ldim}(\mathcal{H})$.

**Definition 2** (Littlestone dimension). We call a sequence of data points $x_1, \cdots, x_{2^d - 1} \in \mathcal{X}$ a shattered tree of depth $d$, when for all labeling $(y_1, \cdots, y_d) \in \{0, 1\}^d$, there exists $h \in \mathcal{H}$ such that for all $t \in [d]$ we have that $h(x_{i_t}) = y_t$, where

$$i_t = 2^{t-1} + \sum_{j=1}^{t-1} y_j 2^{t-1-j}.$$

The Littlestone dimension $\mathbf{Ldim}(\mathcal{H})$ of a hypothesis class $\mathcal{H}$ is the maximal integer $M$ such that there exist a shattered tree of depth $M$. When no finite $M$ exists, we write $\mathbf{Ldim}(\mathcal{H}) = \infty$. We call any hypothesis class with finite $\mathbf{Ldim}(\mathcal{H})$ a Littlestone class.

It's clear from the definition of Littlestone dimension, that no algorithm can have a mistake bound strictly smaller than $\mathbf{Ldim}(\mathcal{H})$. Moreover, there is a deterministic learning algorithm SOA 3 which is guaranteed to make at most $\mathbf{Ldim}(\mathcal{H})$ errors.

**Lemma 3** (Online learnability Littlestone [1988]). *When $\mathbf{Ldim}(\mathcal{H}) < \infty$, no learning algorithm makes errors strictly fewer than $\mathbf{Ldim}(\mathcal{H})$. In addition, Algorithm 3 makes at most $\mathbf{Ldim}(\mathcal{H})$ errors.*

## 2.2 Inductive Inference as Non-uniform Online Learning

Nature's ability of selecting a changing $h_t^*$ adaptively contradicts the principles of inductive inference that the ground-truth should be consistent. To set the stage for inductive inference, the first change we make to classic online learning is requiring Nature to fix a ground-truth hypothesis $h^*$ in advance and sets $y_t = h^*(x_t)$ thereforth. The main criterion we consider in this paper is then hypothesis-wise error bounds depending on $h^*$, besides the dependence on $\mathcal{H}$.

The reason for mainly considering hypothesis-wise guarantees is twofold: (1) it follows the conventions of inductive inference literature (2) uniform guarantee is too strong for the problem of induction, while consistency guarantee is weak in that it's fully posterior and thereby less informative. Throughout this paper, all discussions are made under the above setting, unless otherwise specified (classic/consistency). Denote the number of errors made by a learning algorithm $\mathcal{A}$ when Nature chooses $h$ and presents $x$ to Learner as

$$err_{\mathcal{A}}(h, x) \triangleq \sum_{t=1}^{\infty} |\hat{y}_t - h(x_t)| \in \mathbb{N} \cup \infty,$$

we define non-uniform online learnability:

**Definition 4** (Non-uniform online learnability). We say a hypothesis class $\mathcal{H}$ is non-uniform online learnable, if there exists a deterministic learning algorithm $\mathcal{A}$, such that

$$\exists m : \mathcal{H} \to \mathbb{N}^+, \forall h \in \mathcal{H}, \forall x \in X, err_{\mathcal{A}}(h, x) \leq m(h).$$

In the definition above, Nature is allowed to adaptively choose $x$ (for deterministic Learner, adaptive and oblivious adversaries are equivalent). A stronger yet common assumption is each $x_t$ drawn independently from some unknown $\mu$ fixed by Nature in advance. Let $\mathcal{X}$ be equipped with some fixed separable $\sigma$-algebra, we define non-uniform stochastic online learnability as below.

**Definition 5** (Non-uniform stochastic online learnability). We say a hypothesis class $\mathcal{H}$ is non-uniform stochastic online learnable, if there exists a deterministic learning algorithm $\mathcal{A}$, such that

$$\exists m : \mathcal{H} \to \mathbb{N}^+, \forall h \in \mathcal{H}, \forall \mu, \mathbb{P}_{x \sim \mu^\infty} \left( err_{\mathcal{A}}(h, x) \leq m(h) \right) = 1.$$

**Equivalence to inductive inference**    the non-uniform online learnability notions introduced above are indeed equivalent to the problem of inductive inference, with different rules of observations $\{x_t\}$. Previous results on inductive inference can be written as special cases under our framework with countable-sized hypothesis class $\mathcal{H}$. The most important example is learning computable functions, previously settled by Solomonoff induction, see also Li et al. [2008]. In our language, it concerns a non-uniform online learning problem with $\mathcal{H}$ being a set of (binary) computable functions, which has a countable size. In particular, $\mathcal{H}$ can be the set of possible infinite sequences generated by Turing machines, with $\mathcal{X} = \mathbb{N}^+$ and $x_t = t$.

### 2.3 Agnostic Non-uniform Online Learning

So far we have assumed Nature must choose $y_t$ such that the whole data sequence is consistent with some $h^* \in \mathcal{H}$. However, it's a rather strong assumption that the ground-truth $h^*$ is already included in the hypothesis class in consideration. This motivates the study on the more realistic agnostic learning setting: Nature can select arbitrary $x_t, y_t$ in an oblivious way, where the sequence of $x_t, y_t$ is not necessarily realizable by some hypothesis $h \in \mathcal{H}$.

It's known that for the agnostic online learning setting, even a hypothesis class $\mathcal{H}$ with finite **Ldim**$(\mathcal{H})$ can make $\Omega(\sqrt{T\mathbf{Ldim}(\mathcal{H})})$ errors in expectation for a time horizon $T$ Ben-David et al. [2009], making the finite error criterion less interesting. As a result, we will consider sub-linear dependence on $T$ in the error bound. Agnostic non-uniform online learnability is defined as below.

**Definition 6** (Agnostic non-uniform online learnability). We say a hypothesis class $\mathcal{H}$ is agnostic non-uniform online learnable with rate $r(T)$, if there exists a learning algorithm $\mathcal{A}$, such that

$$\exists m : \mathcal{H} \to \mathbb{N}^+, \forall x \in X, \forall y \in \{0,1\}^\infty, \forall h \in \mathcal{H}, \forall T \in \mathbb{N}^+,$$

$$\mathbb{E}\left[ \sum_{t=1}^{T} 1_{[\hat{y}_t \neq y_t]} - \sum_{t=1}^{T} 1_{[h(x_t) \neq y_t]} \right] \leq m(h) r(T).$$

## 3 Non-uniform Guarantees in the Realizable Setting

In this section we study non-uniform (stochastic) online learnability in the realizable setting. We provide a complete characterization by showing the learnability of $\mathcal{H}$ is equivalent to $\mathcal{H}$ being a countable union of Littlestone classes. This result generalizes previous inductive inference methods such as Solomonoff [1964a] which provide only a sufficient condition $\mathcal{H}$ being countable for this setting. Our analysis begins with the basic uniform case.

**Definition 7** (Uniform online learnability). We say a hypothesis class $\mathcal{H}$ is uniform online learnable, if there exists a deterministic learning algorithm $\mathcal{A}$, such that

$$\exists m \in \mathbb{N}^+, \forall h \in \mathcal{H}, \forall x, err_{\mathcal{A}}(h, x) \leq m.$$

The next lemma shows uniform online learnability is equivalent to classic online learnability.

**Lemma 8.** *$\mathcal{H}$ is uniform online learnable if and only if $\mathcal{H}$ is classic online learnable. In addition, **Ldim**$(\mathcal{H})$ is a tight error bound.*

We are ready to state our main results in the realizable setting. The proof is mainly based on the structural risk minimization (SRM) technique Vapnik and Chervonenkis [1974], Wu and Santhanam [2021], in which our algorithm goes over each $\mathcal{H}_n$ one by one. Intuitively, since each $\mathcal{H}_n$ is uniform online learnable, we can safely exclude the whole class when a certain amount of errors are made by SOA.

**Theorem 9.** *$\mathcal{H}$ is non-uniform online learnable if and only if $\mathcal{H}$ can be written as a countable union of hypothesis classes with finite Littlestone dimension.*

---
**Algorithm 1** Non-uniform Online Learner
---
1: Input: a hypothesis class $\mathcal{H} = \cup_{n \in \mathbb{N}^+} \mathcal{H}_n$ with $d_n = \mathbf{Ldim}(\mathcal{H}_n) < \infty, \forall n \in \mathbb{N}^+$.
2: Initialize a SOA algorithm $\mathcal{A}_n$ for each $\mathcal{H}_n$, and error bounds $e_1, e_2, \cdots$ all equal to 0.
3: **for** $t \in \mathbb{N}^+$ **do**
4:     Observe $x_t$.
5:     Compute $J_t = \mathrm{argmin}_n\{e_n + n\}$.
6:     Predicts $\hat{y}_t$ as $\mathcal{A}_{J_t}$.
7:     Observe the true label $y_t$.
8:     For each $n \in \mathbb{N}^+$, update $e_n = e_n + 1$ if the prediction of $\mathcal{A}_n$ is not $y_t$.
9: **end for**
---

*Proof.* We discuss the two directions respectively. **If:** Suppose $\mathcal{H}$ can be written as a countable union of hypothesis classes $\mathcal{H}_n$ with finite Littlestone dimension $d_n = \mathbf{Ldim}(\mathcal{H}_n)$. We describe a learning algorithm $\mathcal{A}$ (Algorithm 1) which non-uniform online learns $\mathcal{H}$.

Let $\mathcal{A}_n$ be the SOA algorithm which classic online learns $\mathcal{H}_n$ with at most $d_n$ errors. By Lemma 8, $\mathcal{A}_n$ also uniform online learns $\mathcal{H}_n$ with at most $d_n$ errors. Denote $e(t, n)$ as the the number of errors that $\mathcal{A}_n$ would make if it were employed in the first $t - 1$ steps, let

$$J_t = \mathrm{argmin}_n\{e(t, n) + n\},$$

the algorithm $\mathcal{A}$ predicts as $\mathcal{A}_{J_t}$ at time $t$.

Assume the ground-truth hypothesis $h^*$ lies in $\mathcal{H}_k$, then $\mathcal{A}_k$ makes at most $d_k$ errors which is finite. We have that $e(t, k) + k \leq d_k + k$ for any $t$. Therefore, $J_t \leq d_k + k$ for any $t$, meaning $\mathcal{H}$ will only invoke algorithms within $\mathcal{A}_1, \cdots, \mathcal{A}_{d_k+k}$. Furthermore, once an $\mathcal{A}_n$ makes more than $d_k + k$ errors, it will not be chosen by $\mathcal{A}$ anymore, thus $\mathcal{A}$ makes at most $(d_k + k)^2$ errors, and this upper bound only depends on $h^*$ and is independent of the choice of $x$.

**Only if:** Suppose $\mathcal{H}$ is non-uniform online learnable, we would like to show it can be written as a countable union of uniform online learnable hypothesis classes.

For any $h$, denote $d(h)$ as the maximal number of errors made as admitted by the non-uniform online learnability, which is guaranteed to be a finite number. We denote

$$\mathcal{H}_n = \{h : d(h) = n\}.$$

Since $d(h)$ is a finite number for every $h$, we have that $\mathcal{H} = \cup_{n \in \mathbb{N}} \mathcal{H}_n$. Now we only need to prove each $\mathcal{H}_n$ is uniform online learnable. By applying the same algorithm $\mathcal{A}$ which non-uniform online learns $\mathcal{H}$ on $\mathcal{H}_n$, we have that the number of errors made for any $h \in \mathcal{H}_n$ is bounded by $n$, thus $\mathcal{H}_n$ is uniform online learnable with a uniform error bound $n$. $\square$

As a corollary, Example 22 (rational thresholds) shows the existence of a class non-uniform online learnable but not online learnable, demonstrating a separation between the two notions.

**Corollary 10.** *Uniform online learnability $\subsetneq$ non-uniform online learnability.*

Our second result shows that non-uniform online learnability is in fact equivalent to non-uniform stochastic online learnability.

**Theorem 11.** *$\mathcal{H}$ is non-uniform stochastic online learnable $\iff$ it's non-uniform online learnable.*

*Proof.* We only need to discuss the only if direction, since the other direction is immediate from the definitions. It's equivalent to showing the following two are incompatible: (1) $\mathcal{H}$ is non-uniform stochastic online learnable; (2) $\mathcal{H}$ is not non-uniform online learnable.

Suppose (1) and (2) both hold, we would like to derive a contradiction. (1) implies that there exists a learning algorithm $\mathcal{A}$ such that for any $h \in \mathcal{H}$, there is a constant $n(h)$, for any $\mu$, the number of errors made is bounded by $n(h)$ with probability 1.

However, (2) implies for the same algorithm $\mathcal{A}$, there exists a $h \in \mathcal{H}$, such that for any $n$, there is a sequence $x^{(n)} \in X$ that the number of errors is at least $n$ by some finite time $t_n$. Now, pick

$x = x^{(n(h)+1)}$, we define an arbitrary discrete probability measure $\mu$ over $x$ with full support. For example, we can define

$$\mu(x_i^{(n(h)+1)}) = \frac{1}{2^i}, \forall i \in \mathbb{N}^+.$$

There is positive possibility that in the first $t_{(n(h)+1)}$ steps, the sequence is identical to $x$, which incurs $n(h) + 1 > n(h)$ errors, contradicting (1). In particular, the probability is lower bounded by

$$\Pi_{i=1}^{t_{(n(h)+1)}} \frac{1}{2^i} = \frac{1}{2^{\sum_{i=1}^{t_{(n(h)+1)}} i}} \geq \frac{1}{2^{t_{(n(h)+1)}^2}} > 0.$$

$\square$

The two theorems together imply that for both the strongest (adaptive) and weakest (stochastic) Nature's choice on $x$, the characterization for learnability is the same, therein applies to any other setting in between such as stochastic process Hanneke [2021] or dynamically changing environment Wu et al. [2023]. As a result, Theorem 9 and Theorem 11 together imply our main result 1 on inductive inference, now that all natural choices of observation sequences share the same characterization: $\mathcal{H}$ is a countable union of Littlestone classes.

## 4 Non-uniform Guarantees in the Agnostic Setting

The realizable assumption implies a belief that the law of nature already lies in a set of hypotheses known to Learner. This however doesn't match the practical scenarios of scientific progress where theories are usually proven not perfectly correct along with the development of science.

As a result, the more realistic way is to consider the agnostic setting, in which we pose no constraint on how Nature presents $y_t$, with Learner's objective to be performing as good as the best hypothesis $h^*$ from the whole class. We obtain the following regret bound showing that any countable union of Littlestone classes is also learnable in this setting.

**Theorem 12.** *When $\mathcal{H} = \cup_{n \in \mathbb{N}^+} \mathcal{H}_n$ is a countable union of hypothesis classes with finite Littlestone dimension $d_n$, then it's agnostic non-uniform learnable at rate $\tilde{O}(\sqrt{T})$, with expected regret*

$$\mathbb{E}\left[\sum_{t=1}^{T} 1_{[\hat{y}_t \neq y_t]} - \sum_{t=1}^{T} 1_{[h(x_t) \neq y_t]}\right] = \tilde{O}\left((\sqrt{d_n} + \log n)\sqrt{T}\right)$$

*for any $x, y, T$ and $h \in \mathcal{H}_n$. Under $\tilde{O}$ we hide logarithmic dependences except for $n$.*

---

**Algorithm 2** Agnostic Non-uniform Online Learner

---

1: Input: a hypothesis class $\mathcal{H} = \cup_{n \in \mathbb{N}^+} \mathcal{H}_n$ with $d_n = \mathbf{Ldim}(\mathcal{H}_n) < \infty, \forall n \in \mathbb{N}^+$.
2: For each $\mathcal{H}_n$, initialize a Hierarchical FPL instance $\mathcal{A}_n$ with experts being the set of Algorithm 4 instances with $L \leq d_n$ and learning rate $\eta_t = \frac{1}{\sqrt{t}}$. The complexity of an instance with $i_L = j$ is $1 + (d_n + 2) \log j$.
3: Initialize a complexity $k_n = 2(\log n + 1)$ for each expert $\mathcal{A}_n$.
4: **for** $t \in \mathbb{N}^+$ **do**
5:     Choose learning rate $\eta_t = \frac{1}{\sqrt{t}}$.
6:     Sample a random vector $q \sim \exp$, i.e. $\mathbb{P}(q_i = \lambda) = e^{-\lambda}$ for $\lambda \geq 0$ and all $i \in \mathbb{N}^+$.
7:     Output prediction $\hat{y}_t$ of expert $\mathcal{A}_n$ which minimizes

$$\sum_{j=1}^{t-1} 1_{[\mathcal{A}_n(x_j) \neq y_j]} + \frac{k_n - q_n}{\eta_t}.$$

8:     Observe the true label $y_t$.
9: **end for**

---

Furthermore, we show Theorem 12 indeed provides a tight characterization by the following result, that any hypothesis class which can't be written as a countable union of Littlestone classes requires

arbitrarily slow rates, with a different spirit than the typical $\Omega(\sqrt{dT})$ lower bound in Ben-David et al. [2009].

**Proposition 13.** *For any hypothesis class $\mathcal{H}$, if it can't be written as a countable union of Littlestone classes, then it's not agnostic non-uniform online learnable at rate $T^{1-\epsilon}$ for any $\epsilon > 0$.*

On the other hand, it's well-known an $\Omega(\sqrt{T})$ lower bound is inevitable, even for the simplest case with two hypothesis classes differing on only one point.

**Proposition 14.** *As long as $|\mathcal{H}| > 1$, then for any Learner's algorithm and $T \in \mathbb{N}^+$, the expected regret at time $T$ is at least $\Omega(\sqrt{T})$.*

As a result, we reach the following trichotomy that provides a complete characterization of agnostic non-uniform online learnability, including degenerate, typical, and arbitrarily slow rates, which can be seen as an online agnostic analogue to the universal learning trichotomy of Bousquet et al. [2021].

**Theorem 15.** *There are only three possible rates of agnostic non-uniform online learning:*

- *$\mathcal{H}$ is learnable at rate $0 \iff |\mathcal{H}| = 1$.*

- *$\mathcal{H}$ is learnable at rate $\tilde{\Theta}(\sqrt{T}) \iff \mathcal{H}$ is a countable union of Littlestone classes.*

- *$\mathcal{H}$ requires arbitrarily slow rates $\iff \mathcal{H}$ isn't a countable union of Littlestone classes.*

## 5   Non-uniformity Versus Consistency in the Realizable Setting

In addition to hypothesis-wise guarantees, we can also consider the weaker notion of consistency, which needs no uniformity over $x$ for a fixed $h$ and only requires the number of errors to be finite. We will restrict our attention to the realizable setting only, since non-hypothesis-wise guarantees are ill-defined in the agnostic setting.

**Definition 16** (Consistency). A hypothesis class $\mathcal{H}$ is consistent, if there is a deterministic algorithm $\mathcal{A}$, such that $err_{\mathcal{A}}(h, x) < \infty$ for any $(h, x)$ chosen by Nature.

In the classic online learning setting, it was proven by Bousquet et al. [2021] that a hypothesis class $\mathcal{H}$ is consistent, if and only if $\mathcal{H}$ has an infinite-depth shatter tree. Example 21 (integer thresholds) is consistent but not online learnable, showing that uniformity $\neq$ consistency in the classic online learning setting.

To characterize consistency in our setting where Nature must fix $h^*$ beforehand, we introduce the following definition on the size of an infinite (binary) tree, such that besides the set of nodes $V$, each edge is marked with some number $z \in \{0, 1\}$ where left edges are 0 and right edges are 1.

**Definition 17** (Size of infinite tree). Given a hypothesis class $\mathcal{H}$, for any infinite tree, a countable branch $b = (v_1, z_1, v_2, z_2, \cdots)$ where $v_{t+1}$ is the $z_t$-child of $v_t$, is called realizable if there is some $h \in \mathcal{H}$ consistent on $b$. The size of this tree is

$$|B = \{b | b \text{ is realizable}\}| \subset \mathbb{N} \cup \aleph_0 \cup \aleph_1.$$

In addition, we say the tree is full-size if all branches are realizable.

Example 22 (rational thresholds) has an infinite-depth shatter tree yet no $\aleph_1$-size trees. Example 23 (real thresholds), however, has a full-size tree. Clearly, when $\mathcal{H}$ has a full-size tree, it's not consistent, since any deterministic Learner must make no correct predictions on one of the branches. We obtain a stronger necessary condition for consistency by the following result.

**Theorem 18.** *If a hypothesis class $\mathcal{H}$ is consistent in the non-uniform (stochastic) online learning setting, it doesn't have any $\aleph_1$-size tree in which tree nodes belong to $\mathcal{X}$.*

*Proof.* Suppose for contradiction that there exists an $\aleph_1$-size tree $\mathcal{T}$ while $\mathcal{H}$ is consistent at the same time. For each $b \in B$, let $h_b \in \mathcal{H}$ be some hypothesis consistent with $b$. Consider a set $S = \cup_{b \in B} s_b$ of Nature's strategies, where by $s_b$ Nature fixes $h_b$ beforehand and shows the branch $b$ to Learner.

Now, since $\mathcal{H}$ is consistent, there is some Learner's algorithm $\mathcal{A}$ such that it makes $m_b < \infty$ errors when Nature plays $s_b$ for any $b \in B$. Denote $B_m = \{b | m_b = m\}$, we have that $B = \cup_{m \in \mathbb{N}} B_m$.

On the one hand, any countable union of countable sets is still countable, while on the other hand, $|B| = \aleph_1$, therefore there must exists some $m^* < \infty$ such that $|B_{m^*}| = \aleph_1$.

Consider a new hypothesis set $\mathcal{H}^* = \{h \in \mathcal{H} | h \in B_{m^*}\}$. Since it's a subset of $\mathcal{H}$, $\mathcal{A}$ is also consistent on $\mathcal{H}^*$. In particular, the tree $\mathcal{T}$ is still $\aleph_1$-size. Central to our proof is the claim that for any $\aleph_1$-size tree, there exists a node of the tree, such that both induced sub-trees of its children are $\aleph_1$-size.

Suppose for contradiction the claim is false, denote the root as $N_1$, we can recursively define an infinite sequence $N_2, N_3, \cdots$ such that for any $t \geq 2$, $N_t$ is the child of $N_{t-1}$ and $N_t$'s brother's induced sub-tree (denoted as $\mathcal{T}_t$) is at most $\aleph_0$-size. Now the set of realizable branches is countable: besides the single branch defined by the infinite sequence, any other branch falls in one sub-tree $\mathcal{T}_t$, and there are only countably many such $\aleph_0$-size sub-trees $\mathcal{T}_t$. This contradicts the fact that the tree is $\aleph_1$-size and proves the claim.

By repeatedly using the claim, we can find $m^* + 2$ levels of nodes of the tree $\mathcal{T}$

$$v_{(1,1)}, v_{(2,1)}, v_{(2,2)}, \cdots, v_{(m^*+2,1)}, \cdots, v_{(m^*+2,2^{m^*+1})}$$

where $v_{(i,j)}$ is the ancestor of $v_{(i+1,2j-1)}, v_{(i+1,2j)}$, and each $v_{(i,j)}$ induces an $\aleph_1$-size sub-tree. Notice that any node $v_{(i,j)}$ need not to be a $i$-depth node in the tree $\mathcal{T}$, and $v_{(i,j)}$ may be a multi-generation grandfather of $v_{(i+1,2j-1)}, v_{(i+1,2j)}$.

Now, by construction $\mathcal{H}^*$ shatters the $m^* + 2$ levels of nodes, therefore any deterministic algorithm including $\mathcal{A}$ will make at least $m^* + 1$ errors on one of the finite branches (if we see the levels of nodes as a tree). This however contradicts the definition of $B_{m^*}$, which completes the proof. $\square$

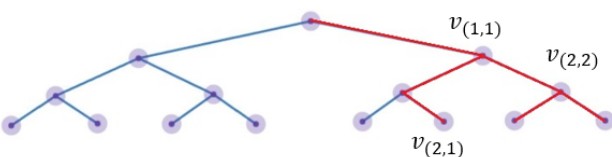

Figure 1: An example of $v_{(i,j)}$. Here the red paths denote realizable branches.

In addition, when $\mu$ is known to Learner in the non-uniform stochastic online learning setting, we strengthen a result of Wu and Santhanam [2021] in which they name consistency as EAS.

**Definition 19** (EAS online learnability). A class $\mathcal{H}$ is eas-online learnable w.r.t distribution $\mu$, if there exists a learning algorithm $\mathcal{A}$ such that for all $h \in \mathcal{H}$ and $x \sim \mu^\infty$

$$\mathbb{P}\left(err_{\mathcal{A}}(h, x) < \infty\right) = 1.$$

They proved that $\mathcal{H}$ is eas-online learnable if and only if $\mathcal{H}$ has a countable effective cover w.r.t $\mu$ (Theorem 7 in Wu and Santhanam [2021]), where $\mathcal{H}'$ is said to be an effective cover if for any $h \in \mathcal{H}$, there exists $h' \in \mathcal{H}'$ such that (we say $h$ is covered by $h'$) $\mu\{h(x) \neq h'(x)\} = 0$. We strengthen this result by showing that in the same setting of Wu and Santhanam [2021], non-uniformity = consistency.

**Proposition 20.** *When $\mu$ is known to Learner, a class $\mathcal{H}$ is non-uniform stochastic online learnable if and only if $\mathcal{H}$ has a countable effective cover w.r.t $\mu$.*

|  | any $x$, varying $h^*$ | any $x$, fixed $h^*$ | unknown $\mu$, fixed $h^*$ | known $\mu$, fixed $h^*$ |
|---|---|---|---|---|
| non-uniform | NA | $\cup_{\aleph_0}$ LS $=$ | $\cup_{\aleph_0}$ LS $\underset{?}{\subsetneq}$ | $\aleph_0$ effective cover |
|  |  | $\cap$ | | $\|$ |
| consistent | no $\infty$ tree $\subsetneq$ | no $\aleph_1$-size tree | ? | $\aleph_0$ effective cover |

Table 1: Summary of learnabilities in the non-uniform realizable setting. LS denotes any Littlestone class. Our results are marked in red. Necessary conditions are marked in blue.

# 6 Discussion

**Summary of Results**   We resolve a long-standing philosophical question, the question of inductive inference, via a novel link to online learning theory. Different from previous results on inductive inference, which only provide a sufficient condition that the hypothesis class has a countable size, we provide a sufficient and necessary condition: inductive inference is possible, if and only if the hypothesis class is a countable union of online learnable classes. We hope this connection offers a fresh perspective that could open new avenues for research in both fields.

From the technical side, we develop a novel framework called non-uniform online learning, where the ground-truth hypothesis is determined in advance and hypothesis-wise error bounds are considered. In both realizable and agnostic settings, we prove that the hypothesis class $\mathcal{H}$ being a countable union of Littlestone classes is the sufficient and necessary condition for learnability. Our results provide a sharp characterization of inductive inference beyond countable-sized realizable hypotheses.

**Philosophical Interpretations**   Our findings reveal an intrinsic trade-off between $d_n$ and $n$ in both the error bound of Theorem 9 the regret bound of Theorem 12, depending on how we make the countable partition $\mathcal{H} = \cup_{n \in \mathbb{N}^+} \mathcal{H}_n$. A finer partition can decrease the Littlestone dimension $d_n$ of the class $\mathcal{H}_n$ that $h$ belongs to for every $h \in \mathcal{H}$, however, at the cost of creating more classes and increasing the dependence on $n$. Such trade-off shows a connection to certain philosophical debates around falsifiability, such as "the mission is to classify truths, not to certify them" Miller [2015].

**Future Directions**   Identifying a sufficient condition for consistency remains an unresolved challenge. We conjecture that having no $\aleph_1$-size tree is a tight characterization. Understanding inclusion relations between different learnabilities, as outlined in the table, is also an important problem. The current work prioritizes error bounds without delving into computational constraints. It's interesting to study non-uniform online learnability with computable learners for this theory's practical applicability, as in Agarwal et al. [2020], Sterkenburg [2022], Hasrati and Ben-David [2023].

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

# A Algorithms from Previous Work

---

**Algorithm 3** Standard Optimal Algorithm (SOA)

---

1: Input: a hypothesis class $\mathcal{H}$.
2: Initialize the active hypothesis set $V_1 = \mathcal{H}$.
3: **for** $t \in \mathbb{N}^+$ **do**
4:    Observe $x_t$.
5:    For $y \in \{0, 1\}$, let $V_t^y = \{h : h(x_t) = y, h \in V_t\}$.
6:    Predict $\hat{y}_t = \mathrm{argmax}_y \mathbf{Ldim}(V_t^y)$.
7:    Observe the true label $y_t$.
8:    Update $V_{t+1} = V_t^{y_t}$ if $y_t \neq \hat{y}_t$. Otherwise set $V_{t+1} = V_t$.
9: **end for**

---

---

**Algorithm 4** Expert$(i_1, \cdots, i_L)$

---

1: Input: a hypothesis class $\mathcal{H}$, indices $i_1 < \cdots < i_L$.
2: Initialize the active hypothesis set $V_1 = \mathcal{H}$.
3: **for** $t \in \mathbb{N}^+$ **do**
4:    Observe $x_t$.
5:    For $y \in \{0, 1\}$, let $V_t^y = \{h : h(x_t) = y, h \in V_t\}$.
6:    Predict $\hat{y}_t = \mathrm{argmax}_y \mathbf{Ldim}(V_t^y)$.
7:    Observe the true label $y_t$.
8:    Update $V_{t+1} = V_t^{y_t}$ if $y_t \neq \hat{y}_t$ and $t \in \{i_1, \cdots, i_L\}$. Otherwise set $V_{t+1} = V_t$.
9: **end for**

---

---

**Algorithm 5** Follow the Perturber Leader (FPL)

---

1: Input: a countable class of experts $E_1, E_2, \cdots$.
2: Initialize a complexity $k_i$ for each expert $E_i$, such that $\sum_{i \in \mathbb{N}^+} e^{-k_i} \leq 1$.
3: **for** $t \in \mathbb{N}^+$ **do**
4:    Choose learning rate $\eta_t$.
5:    Sample a random vector $q \sim \exp$, i.e. $\mathbb{P}(q_i = \lambda) = e^{-\lambda}$ for $\lambda \geq 0$ and all $i \in \mathbb{N}^+$.
6:    Output prediction $\hat{y}_t$ of expert $E_i$ which minimizes

$$\sum_{j=1}^{t-1} 1_{[E_i(x_j) \neq y_j]} + \frac{k_i - q_i}{\eta_t}.$$

7:    Observe the true label $y_t$.
8: **end for**

---

# B Examples

**Example 21.** Consider $\mathcal{X} = \mathbb{N}^+$ and let $\mathcal{H}$ be the set of all threshold functions

$$\mathcal{H} = \{h : h(x) = 1_{[x \geq n]}, n \in \mathbb{N}\}.$$

The Littlestone dimension $\mathbf{Ldim}(\mathcal{H}) = \infty$. However, there is a simple learning algorithm with finite errors, by keeping predicting 0 until the first mistake. It's clear $\mathcal{H}$ is consistent but not uniformly classic online learnable.

**Example 22.** Consider $\mathcal{X} = [0, 1]$, and the class of all rational thresholds

$$\mathcal{H} = \{h : h(x) = 1_{[x \geq c]}, c \in \mathbb{Q}\}.$$

This hypothesis class has infinite Littlestone dimension, thereby not uniformly online learnable. However, We have the countable decomposition $\mathcal{H} = \cup_{c \in [0,1], c \in \mathbb{Q}} \mathcal{H}_c$, where each $\mathcal{H}_c$ is a singleton $\mathcal{H}_c = \{1_{[x \geq c]}\}$. Thus it's non-uniformly online learnable.

**Example 23.** Consider $\mathcal{X} = [0, 1]$, and the class of all real thresholds

$$\mathcal{H} = \{h : h(x) = 1_{[x \geq c]}, c \in \mathbb{R}\}.$$

This hypothesis class is not even EAS online learnable under the uniform distribution Wu and Santhanam [2021], thus not consistent.

In the language of Definition 17, $\mathcal{H}$ has a full-size infinite shatter tree. The naive way of setting the tree nodes as $(\frac{1}{2}, \frac{1}{4}, \frac{3}{4}, \frac{1}{8}, \cdots)$ doesn't give a full-size tree, because the limit of some converging branch can be one of the seen nodes. For example, consider a branch with $x = (\frac{1}{2}, \frac{3}{4}, \frac{5}{8}, \frac{9}{16}, \cdots)$, the limit $\frac{1}{2}$ requires the only possible consistent hypothesis to be $h(x) = 1_{[x \geq \frac{1}{2}]}$, which contradicts the first round error $h(\frac{1}{2}) = 0$.

Instead, we can shrink the window size at each round to prevent the convergence to any node seen. For example, we can half the window size then take the middle point to construct the tree which is a full-size infinite tree

$$\{\frac{1}{2}, \frac{1}{4}, \frac{3}{4}, \frac{3}{16}, \frac{5}{16}, \frac{11}{16}, \frac{13}{16}, \frac{11}{64} \cdots\}$$

## C    Omitted Proofs

We include here the omitted proofs from the main-text, which are either straightforward or standard in literature.

### C.1    Proof of Lemma 3

*Proof.* The first claim is proven by the existence of a shatter tree with depth $\mathbf{Ldim}(\mathcal{H})$. Let $d = \mathbf{Ldim}(\mathcal{H})$ and set $x_t$ to be $x_{i_t}$ as in the definition, then if Nature chooses $y_t = -\hat{y}_t$, Learner makes $d$ mistakes. The existence of consistent hypotheses is guaranteed by the definition of a shatter tree.

For the second claim, it suffices to prove that whenever SOA makes an error, the Littlestone dimension of the active hypothesis set strictly decreases. Assume for the contrary that $\mathbf{Ldim}(V_{t+1}) = \mathbf{Ldim}(V_t)$ but $y_t \neq \hat{y}_t$. The definition of $\hat{y}_t$ implies $\mathbf{Ldim}(V_t) = \mathbf{Ldim}(V_t^1) = \mathbf{Ldim}(V_t^0)$. In this case, we obtain a shatter tree with depth $\mathbf{Ldim}(V_t) + 1$ for $V_t$, which is a contradiction.    $\square$

### C.2    Proof of Lemma 8

*Proof.* We only need to discuss the only if direction, since it's clear from the definition any online learnable problem is automatically uniform online learnable. The claim is reduced to showing uniform online learnability implies online learnability.

Suppose $\mathcal{H}$ is uniform online learnable, then there is a learning algorithm $\mathcal{A}$ and a finite number $d$, such that for any $h \in \mathcal{H}$ and choice of $x \in X$ by Nature, $\mathcal{A}$ makes at most $d$ errors.

We claim that applying the same algorithm $\mathcal{A}$ to the online setting, where Nature is even allowed to adaptively choose consistent $h_t^*$, will also make at most $d$ errors.

We prove by contradiction. Suppose for some $x$ and some time step $T$, the algorithm makes the $(d+1)th$ error at time $T$, while the Nature chooses $h_T^*$ at this time step. Notice that $h_T^*$ is consistent with the whole history of data. The algorithm will also make $d + 1$ errors at time $T$, when Nature fixes $h_T^*$ in advance and uses the same $x$, since it incurs exactly the same history of data. However, this contradicts our assumption on uniform online learnability.    $\square$

### C.3    Proof of Theorem 11

*Proof.* In agnostic online learning, Ben-David et al. [2009] attains an $O(\sqrt{T\mathbf{Ldim}})$ regret bound for Littlestone classes, which however requires $T$ to be known in advance. To get rid of such dependence, we use the FPL algorithm Hutter et al. [2005] for the learning with experts problem with countably many experts and unknown $T$. In particular, we use Algorithm 4 (a subroutine for learning a Littlestone class with known $\mathbf{Ldim}$ and $T$) as the experts for the Hierarchical FPL Algorithm from Hutter et al. [2005], to obtain a regret bound for any Littlestone class in the agnostic setting, without knowing $T$. Having such algorithms for each of $\mathcal{H}_n$ in hand, we then build another hierarchy by

using these algorithm as experts under a meta FPL Algorithm 5, to finally obtain a non-uniform regret bound for every hypothesis in $\mathcal{H}$. We begin with lemmas on the guarantees of these algorithms.

**Lemma 24** (Error Bound of Expert($i_1, \cdots, i_L$) (Algorithm 4), Ben-David et al. [2009])**.** *Let $x$ be any sequence chosen by Nature, for any $T \in \mathbb{N}^+$, we denote $(x_1, y_1), \cdots, (x_T, y_T)$ as the first $T$ rounds of $x, y$. Let $\mathcal{H}$ be a hypothesis class with $\boldsymbol{Ldim}(\mathcal{H}) < \infty$. Then there exists some $L \le \boldsymbol{Ldim}(\mathcal{H})$ and some sequence $1 \le i_1 < \cdots < i_L \le T$, such that Expert($i_1, \cdots, i_L$) makes at most*

$$L + \sum_{t=1}^{T} 1_{[h(x_t) \neq y_t]}$$

*errors on $(x_1, y_1), \cdots, (x_T, y_T)$ for any $h \in \mathcal{H}$.*

**Lemma 25** (Regret Bound of FPL (Algorithm 5), Hutter et al. [2005])**.** *Let $\eta_t = \frac{1}{\sqrt{t}}$ and $k_i$ satisfying $\sum_{i \in \mathbb{N}^+} e^{-k_i} \le 1$, for any $x, y$ and $T \in \mathbb{N}^+$, the regret w.r.t any expert $E_i$ is bounded by*

$$\mathbb{E}\left[\sum_{t=1}^{T} 1_{[\hat{y}_t \neq y_t]}\right] - \sum_{t=1}^{T} 1_{[E_i(x_t) \neq y_t]} \le (k_i + 2)\sqrt{T}.$$

**Lemma 26** (Regret Bound of Hierarchical FPL (Theorem 9 in Hutter et al. [2005]) )**.** *Let $k_i$ satisfying $\sum_{i \in \mathbb{N}^+} e^{-k_i} \le 1$, for any $x, y$ and $T \in \mathbb{N}^+$, there is an algorithm (Hierarchical FPL) such that the regret w.r.t any expert $E_i$ is bounded by*

$$\mathbb{E}\left[\sum_{t=1}^{T} 1_{[\hat{y}_t \neq y_t]}\right] - \sum_{t=1}^{T} 1_{[E_i(x_t) \neq y_t]} \le 2\sqrt{2k_i T}(1 + O(\frac{\log k_i}{\sqrt{k_i}})) = \tilde{O}(\sqrt{k_i T}).$$

Now we explain the details of how we aggregate the algorithms. Fix any $\mathcal{H}_n$ with $\boldsymbol{Ldim}(\mathcal{H}_n) = d_n$, we first extend the result of Ben-David et al. [2009] to infinite horizon.

By Lemma 24, for any $T \in \mathbb{N}^+$, there is an Expert($i_1, \cdots, i_L$) with $i_L \le T$ attaining the claimed error bound. Counting the number of experts with $i_L \le T$, we find there are at most $T^{d_n}$ such experts. In addition, the number of experts with $i_L = T$ is also bounded by $T^{d_n}$.

We specify a way of assigning complexities $k_i$ in Hierarchical FPL. For each expert with $i_L = j$, we assign a complexity $1 + (d_n + 2) \log j$. To see why it's a valid choice, notice

$$\sum_{i \in \mathbb{N}^+} e^{-k_i} \le \sum_{t \in \mathbb{N}^+} t^{d_n} e^{-1-(d_n+2)\log t} \le \sum_{t \in \mathbb{N}^+} \frac{1}{2t^2} \le 0.83.$$

By Lemma 26, it immediately gives an expected regret bound of

$$\tilde{O}(\sqrt{d_n T}),$$

on the Hierarchical FPL instance against Algorithm 4 instances. Now, choose $k_n = 2(\log n + 1)$ for the meta FPL algorithm, we can verify

$$\sum_{n \in \mathbb{N}^+} e^{-k_n} \le \frac{1}{e^2}(1 + \sum_{n \in \mathbb{N}^+} \frac{1}{n^2}) \le \frac{1}{e}.$$

As a result, by Lemma 25, for any $h \in \mathcal{H}_n$, the regret compared with it up to time $T$ is bounded by

$$\tilde{O}(\sqrt{d_n T}) + (2\log n + 4)\sqrt{T} = \tilde{O}\left((\sqrt{d_n} + \log n)\sqrt{T}\right).$$

$\square$

### C.4  Proof of Lemma 24

The proof is a simplified rephrase of Lemma 12 in Ben-David et al. [2009].

*Proof.* Fix any $h \in \mathcal{H}$, we denote $j_1, \cdots, j_k$ be the time steps that $h$ doesn't err. It's equivalent to prove that there exists an $\text{Expert}(i_1, \cdots, i_L)$ such that it makes at most $\textbf{Ldim}(\mathcal{H})$ errors on $j_1, \cdots, j_k$.

Notice that the sequence $(x_{j_1}, y_{j_1}), \cdots, (x_{j_k}, y_{j_k})$ is realizable on $\mathcal{H}$, running SOA on this sequence will incur at most $\textbf{Ldim}(\mathcal{H})$ errors. Choose any sub-sequence $t_1, \cdots, t_L$ of $j_1, \cdots, j_k$ with $L \leq \textbf{Ldim}(\mathcal{H})$, which contains all time steps that SOA makes errors on. The $\text{Expert}(t_1, \cdots, t_L)$ makes the same predictions as SOA on $j_1, \cdots, j_k$, thereby it makes at most $\textbf{Ldim}(\mathcal{H})$ errors on $j_1, \cdots, j_k$, concluding the proof. $\qquad\square$

### C.5   Proof of Proposition 13

Suppose $\mathcal{H}$ is agnostic non-uniform online learnable at rate $T^{1-\epsilon}$ for some $\epsilon > 0$. Define

$$h^* = \text{argmin}_{h \in \mathcal{H}} \sum_{t=1}^{T} 1_{[h(x_t) \neq y_t]}$$

as the best hypothesis in hindsight, the definition of agnostic non-uniform online learnability directly implies that for any $x, y, T$, the expected regret is bounded by

$$\mathbb{E}\left[ \sum_{t=1}^{T} 1_{[\hat{y}_t \neq y_t]} - \sum_{t=1}^{T} 1_{[h^*(x_t) \neq y_t]} \right] \leq m(h^*)T^{1-\epsilon}.$$

We have the partition $\mathcal{H} = \cup_{n \in \mathbb{N}^+} \mathcal{H}_n$, where $\mathcal{H}_n = \{h \in \mathcal{H} : m(h) \leq n\}$. Since $\mathcal{H}$ can't be written as a countable union of Littlestone classes, one of $\mathcal{H}_n$ is not a Littlestone class. Suppose it's $\mathcal{H}_m$, then Learner's expected regret on $\mathcal{H}_m$ is (uniformly) bounded by $mT^{1-\epsilon}$.

However, since it's not a Littlestone class, it has infinite Littlestone dimension. In particular, it has arbitrary depth shatter trees. Fix any shatter tree with depth $T_0 = \lfloor (2m)^{\frac{1}{\epsilon}} \rfloor + 1$, the first integer such that $mT_0^{1-\epsilon} < \frac{T_0}{2}$. Consider the $2^{T_0}$ sequences of $x, y$ defined by this tree. If Nature randomly chooses a sequence among them and presents it to Learner, then Learner has expected loss $\frac{T_0}{2}$ since every prediction is merely a random coin flip. On the other hand, since the tree is shattered, the best hypothesis in hindsight must have loss 0, thus Learner has expected regret at least $\frac{T_0}{2}$. We conclude that it's impossible for Learner to have expected regret at most $mT_0^{1-\epsilon}$ on every sequence since $mT_0^{1-\epsilon} < \frac{T_0}{2}$, leading to the desired contradiction.

### C.6   Proof of Proposition 14

Without loss of generality we can consider the simple problem where $\mathcal{X} = \{0\}$ and $\mathcal{H} = \{h_0(0) = 0, h_1(0) = 1\}$. Nature always selects $x_t = 0$ and generates $y_t$ by iid coin flips, i.e. $y_t = 0$ with probability $\frac{1}{2}$. Clearly, the expected number of errors made by Learner at any time $T$ is $\frac{T}{2}$.

On the other hand, define $z_t = 2y_t - 1$ and $S = \sum_{t=1}^{T} z_t$. We have that $\forall i \neq j$
$$\mathbb{E}[z_i^2] = \mathbb{E}[z_i^4] = \mathbb{E}[z_i^2 z_j^2] = 1, \ \mathbb{E}[z_i z_j] = \mathbb{E}[z_i^3 z_j] = 0.$$
As a result, $\mathbb{E}[S^2] = T$ and $\mathbb{E}[S^4] = T + 3T(T-1) \leq 3T^2$. By the Paley–Zygmund inequality, we have that
$$\mathbb{P}\left( S^2 \geq \frac{T}{4} \right) \geq \frac{3}{16}.$$
Notice the distribution of $S$ is symmetric, we get
$$\mathbb{P}\left( S \geq \frac{\sqrt{T}}{2} \right) \geq \frac{3}{32}.$$

As a result, with probability at least $\frac{3}{32}$, there are $\sqrt{T}$ more 1 than 0 in $\{y_1, \cdots, y_T\}$, and the number of errors made by $h_1$ is at most $\frac{T}{2} - \frac{\sqrt{T}}{2}$. Notice it implies that $\min_{h \in \mathcal{H}} \sum_{t=1}^{T} 1_{[h(x_t) \neq y_t]} \leq \frac{T}{2} - \frac{\sqrt{T}}{2}$, we conclude that
$$\mathbb{E}\left[ \sum_{t=1}^{T} 1_{[\hat{y}_t \neq y_t]} - \min_{h \in \mathcal{H}} \sum_{t=1}^{T} 1_{[h(x_t) \neq y_t]} \right] \geq \frac{3\sqrt{T}}{64}.$$
As a result, $\mathcal{H}$ can't be learnt with rate $o(\sqrt{T})$.

### C.7 Proof of Proposition 20

*Proof.* We only need to discuss the if direction, since non-uniform stochastic online learnability implies EAS online learnability.

Suppose $\mathcal{H}$ has a countable effective cover $\mathcal{H}'$. We claim that with probability 1 on the drawing of $x$, there exists $h' \in \mathcal{H}'$ such that

$$h^*(x_t) = h'(x_t), \forall t \in \mathbb{N}^+. \tag{1}$$

To see this, we fix the $h'$ w.r.t $h^*$ as in the definition of effective cover, then $\mathbb{P}_{x \sim \mu}(h^*(x) \neq h'(x)) = 0$. Since any countable union of zero-measure set still has zero measure, we conclude the proof of the claim.

Since $\mathcal{H}'$ is countable, we list its elements as $h'_1, \cdots, h'_n, \cdots$. We partition $\mathcal{H}$ into $\cup_{n \in \mathbb{N}^+} \mathcal{H}_n$, such that $\mathcal{H}_n$ is the set of $h$ covered by $h'_n$.

Conditioned on the event 1, consider the simple algorithm that outputs the prediction of $h'_n$ which has the smallest index $n$ among all consistent hypotheses in $\mathcal{H}'$. When $h^* \in \mathcal{H}_m$, $h'_m$ is consistent with all $x_t$, then clearly this algorithm makes at most $m$ errors, which is a hypothesis-wise bound as desired. In addition, this happens with probability 1.

$\square$

