# OpenReview forum: "When Is Inductive Inference Possible?"
_NeurIPS.cc/2024/Conference — NeurIPS 2024 spotlight_

### Official Review · Reviewer_Cxri · 2024-06-24

**Soundness:** 3
**Presentation:** 4
**Contribution:** 3
**Rating:** 8
**Confidence:** 3

**Summary:**

The authors *characterize* possible inductive inference by connecting it to online learning.
I find their work extremely interesting!

**Strengths:**

The choice of topic is excellent, delivery is strong.

**Weaknesses:**

See questions.

**Questions:**

Page 1:
Can you please say a few words about the connection of inductive inference to computational complexity and theory of computation?
I have a feeling that some of the papers of Shuichi Hirahara and Mikito Nanashima should be cited.

Page 2:
Can you please elaborate on the comparison of two pseudocode segments?

Page 3:
Line 111:
Notation in math display is confusing :)

Page 4:
Line 124:
Please avoid contractions.

I do not understand Lines 140 -- 147.

Page 5:
Can you please elaborate on Lines 179 -- 181?

Page 6:
Why do you choose $x = x^{n(h) + 1}$?

Page 7:
Can you please elaborate on your notion of Regret?

Page 8:
I am confused by Lines 268 -- 269.

Page 9:
I like the philosophical interpretations!

Please elaborate on future work.
How is your work connected to computational complexity and computational learning theory?

**Limitations:**

None.

---

> ### Author Rebuttal · Authors · 2024-07-31
>
> Thank you for your valuable feedback! We will address your concerns here.
>
> **Connection to computational complexity (page 1)**: good point! Inductive inference (especially Solomonoff's method) is not only linked to learning theory but also to the theory of computation. We will add some discussion on this connection (e.g. the paper "Learning in Pessiland via Inductive Inference" by the authors you mentioned).
>
> **Comparison between pseudocodes (page 2)**: we list them side by side for a straightforward comparison. Only lines 2,6,9 are different. Here lines 2,6 correspond to the difference in protocols: classic online learning allows a changing ground-truth (line 6), while inductive inference requires the ground-truth to be fixed in advance (line 2). Line 9 corresponds to the difference in criteria: classic online learning requires uniform error bounds, while inductive inference allows the bounds to depend on the ground-truth. We will add a caption to briefly explain the differences.
>
> **Lines 140-147 (page 4)**: this paragraph conveys two messages (1) our framework considers general $\mathcal{H}$, while previous work on inductive inference only considers $\mathcal{H}$ with a countable size (2) our framework incorporates different setting on how Nature chooses $x_t$ (adversarial or stochastic), while most previous work only considers a certain rule of nature's choice on $x_t$. By these two, we reach the conclusion that our framework is a general framework of inductive inference, subsuming the previously considered settings. As an example, we cast the problem of learning computable functions under our framework. If you have further questions, please let us know.
>
> **Lines 179-181 (page 5)**: once $A_n$ makes more than $d_k+k$ errors at some time step $t_0$, then for any $t> t_0$, $e(t,n)\ge d_k+k$, and $e(t,n)+n> d_k+k$ since $n\ge 1$. By the definition of how we pick the index $J_t$ (line 5 in Algorithm 1), index $k$ will always be strictly better than index $n$ for any $t> t_0$ because $e(t,k)+k\le  d_k+k$, then our algorithm will never pick $J_t=n$ after $t_0$. Combining it with the fact that only indexes no larger than $d_k+k$ can be possibly invoked, we make at most $(d_k+k)^2$ errors.
>
> **Choice of $x^{n(h)+1}$ (page 6)**: $n(h)+1$ is the smallest number $n$ that guarantees a contradiction, for which we want $n>n(h)$.
>
> **Notion of regret (page 7)**: the regret notion in Theorem 12 is from Definition 6. It's a natural extension of the regret notion in agnostic online learning to the non-uniform setting: we require a uniform dependence on $T$ ($r(T)$ is the same across different $h$), while the overhead $m(h)$ can vary with different $h$.
>
> **Lines 268-269 (page 8)**: Figure 1 can serve as an intuitive explanation. Roughly speaking, we use the claim from lines 261-262 multiple times, to find a sub-tree which is a binary tree with depth $m^*+2$, each node being the root of an $\aleph_1$-size tree. In Figure 1, $v_{(1,1)}$ is the ancestor of the sub-tree, with $v_{(2,1)}$ and $v_{(2,2)}$ being its children (though $v_{(2,1)}$ is not its child in the original tree).
>
> **Future work (page 9)**: as we briefly mentioned, our learning algorithms are inevitably intractable, and we leave a computationally tractable approximate algorithm for future work.
>
> We hope our response has addressed your concerns. If you have further questions, please let us know. Thank you again for your valuable time and insights!

---

> > ### Comment · Reviewer_Cxri · 2024-08-08
> >
> > Thank you! :)

---

### Official Review · Reviewer_4vEY · 2024-07-12

**Soundness:** 4
**Presentation:** 4
**Contribution:** 4
**Rating:** 8
**Confidence:** 3

**Summary:**

This paper establishes a novel link between inductive inference and online learning theory. It introduces a novel non-uniform online learning framework and proves a very interesting result on hypothesis class characterization: the authors show inductive inference is possible if and only if the hypothesis class is a countable union of online learnable classes, irrespective on the observations' distribution.

**Strengths:**

I found this paper is a thorough, beautifully written piece of work that guides the readers intuitively on the series of theoretical results. The necessary and sufficient characterization on hypothesis class when inductive inference is possible is philosophically interesting and would be an interesting addition to the field. The authors presents a wholistic analysis and contextualizing the current work with past literature: through comparison with classic online learning setting, to analysis on agnostic setting and comparison with non-uniformity with consistency. I find this paper would be a valuable addition to the research community.

**Weaknesses:**

The main weakness of this paper is the unclear practical implications of the paper's main result (Theorem 1) on downstream tasks. For example, given one can characterize the hypothesis class as a countable union of online learnable classes - what would it mean to the general machine learning community, e.g., can it offer guidance towards algorithm designs? Could the authors elaborate more on how the potential practical impact of their work?

**Questions:**

(See above)

**Limitations:**

Yes.

---

> ### Author Rebuttal · Authors · 2024-07-31
>
> Thank you for your valuable feedback! We will address your concern here.
>
> **Practical implications**: this work is devoted to a conceptual link between philosophy and learning theory, therefore practical implication is not the primary objective and is beyond the scope of the current work. We believe the new conceptual finding itself is valuable because understanding inductive reasoning is a fundamental problem.
>
> For concrete practical applications, our algorithm has the potential to be made more practical by designing approximate algorithms (as we briefly mentioned in future directions). The original form of Solomonoff induction is also known to be intractbale, but later works built approximate versions of Solomonoff induction.
>
> Some recent works study the practical use of inductive inference in large language models (for example, "Learning Universal Predictors" by Grau-Moya et al 2024), and we believe our algorithms can be useful in practice by a similar approach, for understanding if large language models implicitly learn such Bayesian algorithms during inference. We leave these as future research directions.

---

> > ### Comment · Reviewer_4vEY · 2024-08-10
> >
> > Thank you for the response!

---

### Official Review · Reviewer_y4NC · 2024-07-13

**Soundness:** 3
**Presentation:** 2
**Contribution:** 3
**Rating:** 5
**Confidence:** 2

**Summary:**

This paper studies the non-uniform online learning problem, where error bounds can depend on the hypothesis rather than being uniform across hypotheses. In particular, the paper derives theoretical results on (i) conditions for non-uniform online learnability; (ii) regret bounds when the true hypothesis lies outside of the hypothesis class; and (iii) necessary condition for consistency, where error bounds can additionally depend on the data sequence.

**Strengths:**

The paper comprehensively studies the non-uniform online learning problem. The theoretical results seem sound, drawing upon existing uniform learnability results. I would be interested in the opinion of a learning theory expert on the significance of the results.

**Weaknesses:**

Given that a major claimed contribution of the work is the conceptual link between inductive inference and non-uniform online learning, I would have expected a more formal description of the equivalence (e.g. side-by-side mathematical definitions, with references), and a more detailed contextualization of the existing works in both areas and how they are subsumed by the proposed framework. For example, what are the ‘different rules of observations’? It seems to me that the informal statement of inductive inference (finite number of errors given hypothesis) is naturally formulated as Definition 4.

In terms of presentation, I feel that some examples of hypothesis classes and learning problems in the main paper would be helpful in making the motivation of the paper more accessible; along the lines of the Examples in the Appendix but perhaps more diverse. See Qns below for some specifics

**Questions:**

- Do the authors have an example of a hypothesis class which is a countable union of Littlestone classes but not countable? Examples 21 and 22 are both countable so don’t illustrate the value of the new result.

- Similarly, for the consistency definition, what is an example of a class that is not non-uniform learnable but consistent?

**Limitations:**

I do not have sufficient expertise in this area to assess the limitations of the theoretical work.

---

> ### Author Rebuttal · Authors · 2024-07-31
>
> Thank you for your valuable feedback! We will address your concerns here.
>
> **Significance of the results**: we briefly summarize our contributions here (1) we give a sufficient and necessary condition for inductive inference, a fundamental problem in philosophy, while previous works only provided sufficient conditions (2) to solve this problem, we introduce a new framework called non-uniform online learning, which is closely connected to other learnability notions and can be of independent interest. We hope this addresses your concern on the significance of our results, and we are happy to answer further questions if any.
>
> **More explanation on equivalence**: the equivalence is discussed in lines 140-147. Since our framework is a strictly more general form than previously considered inductive inference by removing constraints on $\mathcal{H}$ and $x_t$, we believe it's unnecessary to have another side-by-side mathematical definitions (as opposed to line 67). We give a representative example of how the problem of learning computable functions is subsumed by our framework in lines 143-147.
>
> **Different rules of observations**: it refers to different Nature's choices of observations $x_t$. Here Definition 4 formulates the adversarial case while Definition 5 handles the stochastic case. By Theorem 9 and Theorem 11, we prove that the two cases share the same sufficient and necessary condition, which implies this condition is also sufficient and necessary for any "rule of observation" between the adversarial and stochastic cases. This corresponds to "different rules of observations" in line 141.
>
> If our explanation on the equivalence still feels unclear to you, we are happy to hear from your further suggestions and revise our writing correspondingly.
>
> **More examples**: we put the examples in the appendix due to space limit, we will add some key examples back to the main paper for readability as you suggested. Below we answer your questions.
>
> **A countable union of Littlestone classes but not countable**: we give a stronger example which is a Littlestone class with an uncountable size. Consider the set of indicator functions on $[0,1]$, i.e. $\{f_c| f_c(x)=1_{x=c}, c\in [0,1]\}$. The size of this hypothesis class is uncountable because $[0,1]$ has the same cardinality as $\mathbb{R}$. This class itself has Littlestone dimension 1. A naive algorithm that makes at most one error on this class is the following: always predict $y_t=0$ until making the first error on some $x_t=c$, then the ground-truth $h^*$ is identified as $f_c$ and we predict w.r.t. $f_c$ afterwards.
>
> **Not non-uniform learnable but consistent**: this is left as an open question, as shown in Table 1. Currently we only know non-uniform learnablity is a subset of consistency and we obtained a new necessary condition for consistency (Theorem 18). It's unclear whether consistency can be separated from non-uniform learnablity or not.
>
> If our response has addressed your concerns, please consider reevaluate our paper. If you have further questions, please let us know. Thank you again for your valuable time and insights!

---

> > ### Comment · Reviewer_y4NC · 2024-08-12
> >
> > Thank you for clarifying that the non-uniform online framework is a novel contribution. I've updated my score accordingly.

---

> > > ### Author Response · Authors · 2024-08-13
> > >
> > > Dear reviewer,
> > >
> > > Thank you for your feedback and appreciation of our results!
> > >
> > > We would like to emphasize that the main contribution of this work is answering the philosophy question "when is inductive inference possible?". The new non-uniform online learning framework we introduced only serves as the tool to solve this question, thereby we consider it as a technical contribution. We hope this message is clearly conveyed to you.
> > >
> > > **Main contribution**: we study inductive inference, a basic problem in philosophy, which is not only crucial to understanding human reasoning, but also inspired pioneer works in learning theory (e.g. "Occam's Razor" [1]). Different from previous works which only considered countable-sized hypothesis classes and thereby provided sufficient conditions, we provide a **necessary and sufficient** condition for inductive inference. This condition is proven tight across various settings (adversarial/stochastic, realizable/agnostic).
> > >
> > > **Technical contribution**: our results are proven via the introduction of a **new learning framework**, non-uniform online learning. This framework is not only a strictly more general form of (previously considered) inductive inference [2], but also a natural combination between classic online learning [3] and non-uniform PAC learning [4]. It's closely connected to other learnability notions (e.g. [5],[6]) and can be of independent interest for future research.
> > >
> > > In conclusion, we believe our results make a solid contribution to both the fields of philosophy and learning theory. We are more than happy to provide further clarification or explanation promptly if required.
> > >
> > > [1] Occam’s Razor, Blumer et al, Information processing letters 1987
> > >
> > > [2] Language identification in the limit, EM Gold, Information and control 1967
> > >
> > > [3] Learning quickly when irrelevant attributes abound: A new linear-threshold algorithm, Nick Littlestone, Machine learning 1988
> > >
> > > [4] Nonuniform learnability, Benedek and Itai, Automata, Languages and Programming 1988
> > >
> > > [5] A theory of universal learning, Bousquet et al, STOC 2021
> > >
> > > [6] Non-uniform consistency of online learning with random sampling, Wu and Santhanam, ALT 2021

---

### Decision · Program_Chairs · 2024-09-25

**Decision:**

Accept (spotlight)

**Comment:**

This paper gives a full characterization of when inductive inference is possible, in both realizable and agnostic settings, strengthening past sufficient conditions with an elegant necessary and sufficient condition. The achievable algorithms, however, are inherently intractable. The technical contributions toward building this theory are novel and the paper is presented very methodically (albeit not for the uninitiated in learning theory, with many concepts assumed known, e.g., the SOA algorithm.) Reviewers appreciated the work and requested a few clarifications that ought to be part of the revision of the paper.